# Influence of Production Factors on Beef Primal Tissue Composition

**DOI:** 10.3390/foods11040518

**Published:** 2022-02-11

**Authors:** Vipasha Sood, Argenis Rodas-González, Stephanie Lam, Óscar López-Campos, Jose Segura, Timothy Schwinghamer, Michael Dugan, John Basarab, Jennifer Aalhus, Manuel Juárez

**Affiliations:** 1Department of Food and Human Nutritional Science, Faculty of Agricultural and Food Sciences, University of Manitoba, Winnipeg, MB R3T 2N2, Canada; soodv@myumanitoba.ca; 2Department of Animal Science, Faculty of Agricultural and Food Sciences, University of Manitoba, Winnipeg, MB R3T 2N2, Canada; Argenis.RodasGonzalez@umanitoba.ca; 3Lacombe Research and Development Centre, Agriculture and Agri-Food Canada, Lacombe, AB T4L 1W1, Canada; stephanie.lam@agr.gc.ca (S.L.); oscar.lopezcampos@agr.gc.ca (Ó.L.-C.); jose.seguraplaza@agr.gc.ca (J.S.); mike.dugan@agr.gc.ca (M.D.); jennifer.aalhus@agr.gc.ca (J.A.); 4Lethbridge Research and Development Centre, Agriculture and Agri-Food Canada, Lethbridge, AB T1J 4B1, Canada; timothy.schwinghamer@canada.ca; 5Agricultural, Food and Nutritional Sciences, Faculty of Agricultural, Life and Environmental Sciences, University of Alberta, Edmonton, AB T6G 2R3, Canada; John.Basarab@gov.ab.ca

**Keywords:** breed, calf-fed, cutout, implants, primal cut, yearling-fed

## Abstract

This study used 1076 crossbred steers to evaluate the effects of calf-fed and yearling-fed beef production systems, implant strategies (with and without implants), and their interactions on the primal tissue composition (lean and fat components) of individual primal cuts using complete carcass dissection data. The results indicate that production system × implant interactions affected loin and rib primal weight percentages as well as marbling (*p* < 0.05) but did not affect the dissectible lean and fat contents of the individual primal cut (*p* > 0.05). Implants increased lean and decreased fat tissue contents of primal cut; however, the production system only affected lean content in the loin (*p* < 0.05) and fat content in the loin, round, and rib (*p* < 0.05). Redundancy analysis revealed a strong association between Angus breed percentage and marbling, as well as between Simmental breed percentage and multiple primal lean traits. Response surface regression models explained less variability in the tissue composition traits in calf-fed compared with yearling-fed animals, suggesting the need for further exploration using genomic studies.

## 1. Introduction

Currently, beef cattle management practices and genetic selection programs aim to improve the lean yield percentage and marbling score measured at a single location in the carcass (ribeye grading site), which are assessed using linear measurements and subjective scores, respectively. Prior studies have explored the interrelationship between primal cut weights in beef cattle [1,2,3,4]. Owing to, in part, the difficulty and cost of producing abundant and accurate phenotypic data from beef carcasses, a tissue composition (lean, fat, and bone) study that assesses primal cuts from the whole carcass has not been performed on a representative sample that is characterized by the breed composition of a typical Western Canadian commercial beef cattle herd, until now.

The characterization of the tissue composition of individual primal cuts will enable information-based decision-making regarding selection and production systems (PS) to increase edible lean meat yield and reduce the fat content of beef carcasses. Fat production is metabolically inefficient and leads to increased feed costs, greenhouse gases emissions, and food waste; therefore, information-based decision-making is hypothesized to improve production economics and reduce the environmental impacts of the beef industry [5]. At the same time, new grading technologies, such as dual-energy X-ray absorptiometry, are being implemented in commercial plants, not only to accurately measure total carcass and individual primal composition [6,7] but also to enable automation in the slaughter and processing areas. The implementation of such new technologies by the North American beef industry will produce more relevant information regarding primal composition and could lead to differential payment systems.

Response surface modeling of production factors that contribute to primal composition can estimate the optimal proportions of lean, fat, and bone for improved profitability and sustainability. The effects of production factors such as PS, hormonal implants (IMP), and breed on overall carcass composition have been well studied for beef cattle. The implementation of different PS has been studied, such as calf-fed (CF) and yearling-fed (YF) systems, with YF known to produce heavier carcasses [8,9,10] and affect estimated yield and overall profitability without significantly affecting quality grade [11,12,13]. The effects of using anabolic implants, such as estradiol benzoate, progesterone, and trenbolone acetate to improve protein deposition [10], growth rate, and muscle growth [14] in cattle are also well established, as well as some potential side effects on meat quality [15]. Their effect on carcass and meat quality traits such as marbling, however, remains contradictory [16,17,18,19], which may be due to interactions with other factors such as breed composition [20]. Few studies have modeled the interaction between these production factors as a contributor to the variance of carcass merit traits [10,20,21], but the variation in tissue composition of individual primal cut due to the interactions among these factors remains unexplored.

Therefore, the objective of this study was to evaluate the effects of production factors, namely production systems (CF or YF), implant use, and their interactions along with breed composition on individual beef primal tissue composition (i.e., lean, fat, and bone).

## 2. Materials and Methods

### 2.1. Live Animals and Slaughter

Experimental conditions were approved by the Agriculture and Agri-Food Canada Lacombe Research and Development Centre (AAFC-LRDC) Animal Care Committee (approval number 201705), in compliance with the principles and guidelines of the Canadian Council on Animal Care [22]. This study used 1076 steers with a breed composition representative of the Western Canadian commercial cattle population [23,24] that were selected from the comprehensive AAFC-LRDC Phenomics database (5000+ animals, 1500+ full cutouts). The breed parentage composition of this population corresponded to 42 Angus (AN), 13 Simmental (SM), and 9 Hereford (HH) bulls used as the sires. The remaining breed background was due to the genetics of the dams, which comprised of the same breeds, i.e., AN, SM, HH, and small percentages of some other breeds. The steers were finished on a commercial diet (i.e., containing 80–90% barley grain on a dry matter basis) under commercial management conditions at the AAFC-LRDC. The steers were raised under two different production systems (CF and YF). CF represents calves (usually heavier calves or large-framed calves with a higher percentage of continental breeds) that were weaned at 6–7 months of age, adjusted to a high grain diet over 1–2 months, and then fed a high-concentrate diet until slaughtered at 14.73 ± 1.44 months of age. YF represents calves (usually lighter calves or smaller-framed calves with a higher percentage of British breeds) that were weaned at 6–7 months of age, fed a backgrounding diet (higher forage content) for 5–6 months, and then fed a high-concentrate diet until slaughter at 20.80 ± 2.51 months of age. The PS and implant groups have been described in detail in previous publications [11,21,25,26]. Steers also belonged to two different growth implant groups: animals with implants (IMP) and animals without implants (no IMP). The distribution of the animals per treatment is depicted in Table 1.

Monthly weights and ultrasound backfat thickness collected via ultrasound (Aloka 500 V diagnostic real-time ultrasound machine, 17 cm 3.5-Mhz linear array transducer; Overseas Monitor Corporation Ltd., Richmond, BC, Canada) were used to allocate animals to different treatments and slaughter dates based on a visual appraisal of body weight and body fatness. Then, the animals were slaughtered and processed at the AAFC-LRDC federally inspected abattoir. At the time of slaughter, final live weights and slaughter dates were recorded, and animals were stunned, exsanguinated, and dressed in a simulated commercial manner.

The data included pedigree, dam and sire breed composition (AN, HH and SM), birth and weaning weights and dates, and full carcass evaluation information, along with complete primal cut composition, collected over 10+ years as a part of multiple studies focused on carcass composition.

### 2.2. Carcass Evaluation and Fabrication

Following slaughter, carcasses were dressed, split, and hot carcass side weights were recorded. Dressing percentage and commercial weights (weight of carcass sides after removal of head, hooves, and viscera) were also calculated. After chilling at 2 °C for 48 h, left and right carcass sides were weighed to determine cooler shrink loss. Both carcass sides were then knife-ribbed between the 12th and 13th ribs. After 20 min of atmospheric exposure, full Canadian grade data were collected by a certified grader from the Canadian Beef Grading Agency. The grading data included the name of the grader, pH and temperature of the carcass, fat thickness (fat thickness over the rib at one-quarter, one-half, and three-quarters position from the spinous process), grade fat (minimum fat thickness over the rib in 4th quadrant from the spinous process), ribeye area (REA; in cm^2^ of the longissimus thoracis), muscle score, quality grade, estimated total lean meat yield calculated according to the Canadian beef grading equations [27], and marbling scores, which were assessed subjectively using pictorial standards as reference points [28]. Intramuscular fat (IMF) was also measured in a longissimus muscle sample using either Soxtec extraction or SMART trac fat analysis [29].

Left carcass sides were fabricated manually into primal cuts with carcass break points identified following the Institutional Meat Purchase Specifications (IMPS) for Fresh Beef Products, Series 100 [30]. The primals collected from the left fabricated carcass side were the chuck (IMPS #113), rib (IMPS #103), brisket (IMPS #118), flank (IMPS #193, non-trimmed), shank (IMPS #117), loin (IMPS #172A), round (IMPS #158A), and plate (IMPS #121). Each whole primal was first weighed, then dissected into fat (subcutaneous, intermuscular, and body cavity fat), lean, and bone separately for each primal, and weighed manually and/or estimated using dual-energy X-ray absorptiometry (DEXA). The data obtained included the fabrication or cutout methodology, individual whole primal cut weights, and individual lean, fat, and bone components of each primal as well as the whole carcass. The results from the dissection of the primals were also transformed to proportional tissue weights within individual primals and total proportional tissue weights within the carcass.

### 2.3. Statistical Analysis

Data were analyzed using the GLIMMIX procedure of the statistical analysis software SAS version 9.4 (SAS Institute Inc., Cary, NC, USA). The fixed effects included the production system (CF and YF), growth implant group (IMP and no IMP), and the interaction (PS × IMP), with individual breed composition (% AN, HH, and SM) used as covariates. A random effect, i.e., the original project that each animal belonged to, was included in the mixed model. In addition, depending on the variable under study, other random effects (such as ‘cutout methodology’ for cutout parameters, ‘grader’ for grading traits, and ‘fat determination method’ for objective fat measurements) were included in the model. Models were compared to select the distribution with the lowest value of the Bayesian information criterion [31]. Breed percentage covariates were tested for significance in the model, and non-significant breed variables were removed (Appendix A). To assess the correlation between the response and explanatory variables, including breed percentage, multivariate redundancy analyses (RDA) were performed using the rda function from the vegan package in R [32,33], in which the categorical variables were transformed into binary indicator variables using the dummyVars function from the caret package in R [34]. Response surface regression models were performed using the RSREG procedure (SAS 9.4, SAS Institute Inc., Cary, NC, USA).

## 3. Results and Discussion

The effects of least squares means of PS, IMP, and the PS×IMP interaction on carcass characteristics, i.e., commercial weight, REA, grade fat, marbling, muscle score, fat class, dressing, intramuscular fat, and estimated yield are presented in Table 2.

Carcasses from YF steers were characterized by higher slaughter weight, REA, muscle scores, dressing percentage, grade fat, fat class, and IMF that were greater than CF steers. While the mean commercial weights of CF steers were significantly lower than YF steers, the SAS PROC GLIMMIX estimated mean of lean yield percentage from CF steer carcasses was greater than YF steers (*p* < 0.01). Similar results were reported in a study [10] in which the carcass weight, dressing percentages, grade fat, and REA of YF steers were greater than CF steers, but the overall estimated lean yield percentage was greater in carcasses from CF steers than the YF steers. The observed lower yield in YF steers aligns with the general allometric growth pattern of cattle, which suggests that fat accumulation increases sharply after lean production begins to subside [35]. Therefore, it is hypothesized that, although in older animals (i.e., YF steers), lean fractions were heavier than CF steers, the higher proportional gain in fat resulted in lower lean meat yield.

The implanted group was characterized by means of slaughter weight, REA, muscle score, and dressing percentage that were statistically significantly greater than the non-implanted group (*p* < 0.01), and the mean of IMF from implanted animals was less than animals without implants (*p* < 0.01). Implants did not affect grade fat, fat class, and estimated yield (*p* > 0.05). Interactive effects between the production system and implants were statistically significant for marbling, which was higher in the no IMP YF compared with IMP CF animals (*p* < 0.05). Previous studies have also reported that marbling scores were affected by PS and IMP individually but did not find any significant interactions [10,11]. The current data indicates that carcasses from the no IMP YF animals were characterized by marbling that was greater than the other experimental groups (*p* < 0.05), which can be attributed to the effects of hormonal implants that can increase lean muscle mass, which in turn decrease the proportion of fat in the animal [36]. YF animals were characterized by more marbled meat, attributed to the development of fat tissues in the later stages of growth, compared with CF animals.

The main effects of PS and IMP were on the primal weight percentages synergistically (Table 3). The interactive effects of PS and IMP were found to be statistically significant on the loin and rib (*p* < 0.05). Mean loin weight percentage was found to be the highest in implanted YF animals and the lowest in no IMP YF animals, whereas mean rib percentage was the highest for implanted CF and lowest in IMP YF steers. These results suggest that although there was an overall increase in the primal weight in YF, implants increased the weight percentage in loin and decreased weight percentage in rib. This indicates that there could be a difference proportional to lean and fat tissue gain in the two primals. The YF production system significantly increased the cut weight percentages of chuck, brisket, and plate (*p* < 0.05), decreased round, flank, and shank (*p* < 0.05), and did not statistically significantly affect loin and rib weight percentages (*p* > 0.05). Implants increased loin and chuck weight percentages (*p* < 0.05) and decreased flank, plate, and shank weight percentages (*p* < 0.05) but did not affect the weights of round, brisket, and rib primal cuts to a statistically significant degree.

Regarding tissue composition (Table 4), YF steers were characterized by lower lean percentage in the loin and higher in the shank, while the PS did not affect other primal cuts or the total lean meat yield percentage from the carcass. The PS affected the loin lean content (*p* < 0.05); however, the other primals were not affected to a statistically significant degree. Implants resulted in greater lean content in all the primal cuts, as well as in the whole carcass. This is because implanted cattle continue to deposit protein, increasing their body size relative to non-implanted cattle at the same body composition [37,38]. The PS × IMP interaction effects (*p* > 0.05) were not statistically significant for the lean fraction of the primal cuts.

The YF production system showed greater dissectible fat in the round, loin, rib, and carcass (*p* < 0.05), while IMP, as expected, resulted in lower fat content in the carcass (*p* < 0.01) and most primal cuts (*p* < 0.05). The shank was affected by the PS × IMP interaction in which CF steers without IMP and YF steers with implant were characterized by means of fat that were greater than implanted CF steers (*p* < 0.05). Previously, a study has reported the statistical significance of the backgrounding effect on rib weight but not on dissectible rib fat [39].

As described earlier, age at slaughter is a distinctive component of the PS, and it affects carcass composition. The findings of a study explained that when animals are slaughtered within the normal slaughter age range, muscle percentage decreases and fat percentage increases [40]. Therefore, with reduced slaughter age, as observed in CF steers, the animals can be in the muscle growth phase rather than the fat accumulation phase, and the fat in the major primals such as the loin, round, rib, and the whole carcass, is significantly lower compared with YF steers.

The proportion of variance explained by each production system for the primal cuts and tissue components is shown in Table 5. The variance for the cut weight percentage of the primals varied greatly for the two production systems and ranged from 14% to 43% for CF, whereas YF explained more variance for the whole primal cut weight and ranged from 31% to 93%. The variability in the lean and fat fractions in each primal was also better explained for YF than for CF steers, being about two-fold for most primals for the former. This could be attributed to the plateauing of tissue growth in older animals, whereas in younger animals, a proportion of the variance could still be unexpressed by phenotypic traits. YF explained a higher proportion of variance for primal lean than the fat content, whereas it was similar for both tissue contents in CF animals. This may be owing to the effect of other factors in the YF production system that may interact with the lean-to-fat ratio. The proportion of variance unexplained by the model is hypothesized to be attributable to genetic factors, and future exploration of genetic variation and primal cut composition will elucidate the hypothetical factors.

Figure 1 and Figure 2 are interpreted as correlation triplots, and the angles between response and explanatory variables and between explanatory variables reflect their correlations [41]. In Figure 1 (RDA1 vs. RDA2) and Figure 2 (RDA1 vs. RDA3), British breeds AN and HH are grouped together on the left of RDA1 with the slaughter age and YF. In contrast, continental breeds, SM grouped on the right side of RDA1 with IMP. In Figure 1, AN, HH, and animals with no IMP show a strong and opposite effect along RDA2 to SS and animals with IMP, indicating a characteristic of being fattier in the former and leaner in the latter. This is further explained by the grouping observed in RDA1, in which marbling, grade fat, fat class, and total fat, as well as individual primal dissectible fat contents, are associated with the breeds AN and HH and older animals under YF without IMP. Conversely, estimated lean yield, total lean, and all primal lean contents are associated with SM and younger animals under CF PS with implants. Moreover, muscle score, REA, and loin primal weight percentage vectors were observed to be in the same direction of slaughter age and YF, suggesting an increase in these traits with increasing age, which is similar to the results obtained from the GLIMMIX procedure presented in Table 1. Negative associations of age at slaughter and YF with loin lean were also observed along RDA1, which was also supported by the GLIMMIX analysis results reported in Table 2.

In Figure 2, RDA1 and RDA3 describe the coordinates of breeds in quadrants where correlated traits colocate. Based on the correlation triplot (Figure 2), AN breed was correlated with marbling. This correlation has been well reported in the literature, in which AN or AN crosses were reported to have higher marbling than HH and continental breeds [39,42]. The AN breed is also associated with the fat content of fattier cuts such as brisket. On the other hand, HH and YF appeared to be highly correlated based on the small angle between the associated vectors in the correlation triplot. HH was also associated with slaughter age, grade fat, fat class, and other primal fat components like loin, round, and plate fat. On the other hand, SM is associated with CF and loin lean. The estimated yield showed a negative association with age at slaughter, grade fat, fat class, and primal dissectible fat contents but associated positively with the lean content of the primals. Loin lean, separated from the rest of the lean traits, revealed that lean and fat tissue composition can greatly vary among different primal cuts, which suggests that predictions of the lean and fat tissue composition of one primal cut are not necessarily representative of the composition of other primal cuts.

## 4. Conclusions

The results of the current study suggest that PS, IMP, and PS × IMP affected the primal weights as well as the lean and fat content of the primals differently. We found an interactive effect of PS × IMP on the loin weight; however, it was not significant of lean and fat fractions of the loin. On the other hand, PS significantly affected all primal cuts except loin and rib and had a significant effect on the loin lean as well as rib and loin fat content. It is also evident from the RDA plots that loin lean did not closely associate with the lean content of other major primals. Therefore, the whole carcass estimates and predictions made using the data collected on a single grading site may not be completely reliable. These results could have a potential effect on the value of grading carcasses using just one primal, which is influenced differently compared with the other primals. The effect of breed composition is also unclear, justifying a need for a deeper investigation using genomic studies to better understand the influence of genetics on beef primal composition.

## Figures and Tables

**Figure 1 foods-11-00518-f001:**
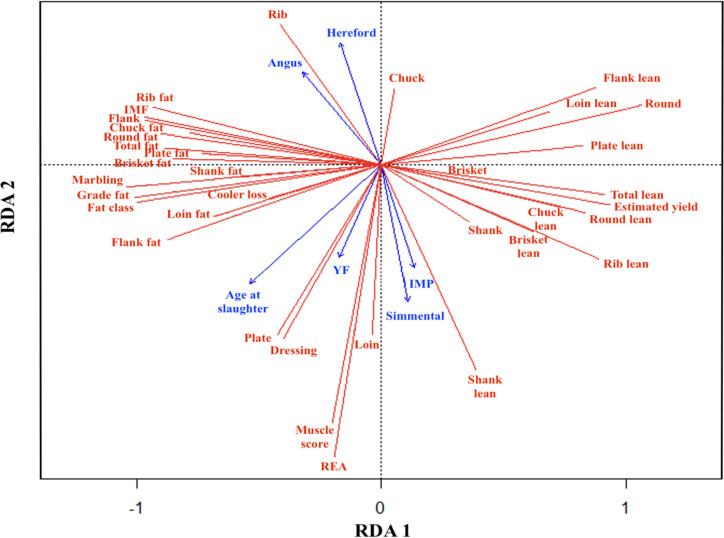
Correlation triplots (scaling = 2) RDA1 vs. RDA2 explaining relationship between independent and dependent variables used in this study: IMF, intramuscular fat; REA, ribeye area; YF, yearling-fed production system; IMP, implants.

**Figure 2 foods-11-00518-f002:**
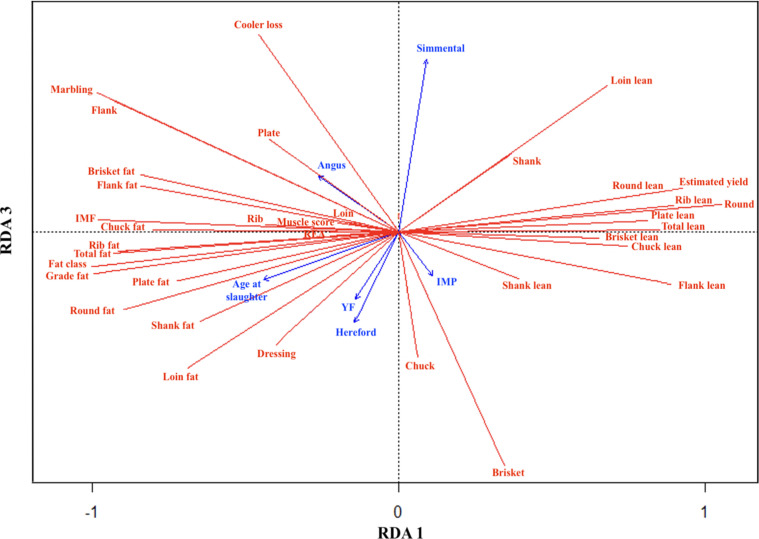
Correlation triplots (scaling = 2) RDA1 vs. RDA3 explaining relationship between independent and dependent variables used in this study: IMF, intramuscular fat; REA, ribeye area; YF, yearling-fed production system; IMP, implants.

**Table 1 foods-11-00518-t001:** Distribution of animals within each treatment combinations of production system (CF and YF) and use of implants (No IMP and IMP).

	CF	YF	Total
IMP	256	255	511
No IMP	317	248	565
Total	573	503	1076

**Table 2 foods-11-00518-t002:** Effects of production system (PS), implants (IMP), and their interactions (PS × IMP) on carcass characteristics (mean ± standard error).

	CF		YF		*p*-Value		
	No IMP	IMP	No IMP	IMP	PS	IMP	PS × IMP
**Commercial** **Weight (kg)**	326 ± 10.8	347 ± 11.5	382 ± 12.7	409 ± 13.6	<0.01	<0.01	0.66
**REA (cm^2^) ^1^**	81.7 ± 1.76	86.8 ± 1.87	87.2 ± 1.89	90.2 ± 1.96	<0.01	<0.01	0.07
**Grade fat (mm)**	11.4 ± 1.35	11.6 ± 1.39	13.2 ± 1.58	13.5 ± 1.62	<0.01	0.29	0.94
**Marbling ^2^**	406 ^b^ ± 25.5	384 ^c^ ± 24.3	444 ^a^ ± 28.0	398 ^b^ ± 25.2	<0.01	<0.01	0.01
**Muscle score**	2.40 ± 0.06	2.96 ± 0.05	2.92 ± 0.05	3.20 ± 0.04	<0.01	<0.01	0.10
**Fat class**	4.50 ± 0.04	4.50 ± 0.04	5.40 ± 0.04	5.40 ± 0.04	<0.01	0.61	0.72
**Dressing (%)**	57.8 ± 0.37	58.4 ± 0.37	58.4 ± 0.37	58.8 ± 0.37	<0.01	<0.01	0.32
**Estimated yield (%)**	57.1 ± 0.95	57.8 ± 0.96	56.3 ± 0.96	56.3 ± 0.97	<0.01	0.11	0.13
**Fat (%) ^3^**	4.13 ± 0.64	3.51 ± 0.55	4.95 ± 0.76	3.88 ± 0.60	<0.01	<0.01	0.10

^a, b, c^ Means with different superscripts were significantly different (*p* < 0.05); ^1^ REA, ribeye area; ^2^ marbling scores range from practically devoid (100–199), traces (200–299), slight (300–399), small (400–499), modest (500–599), moderate (600–699), to slightly abundant (700–799); ^3^ fat (%) refers to the objective measurement of intramuscular fat using analytical techniques; CF, calf-fed production system; YF, yearling-fed production system; IMP, implanted; No IMP, non-implanted.

**Table 3 foods-11-00518-t003:** Effects of production system (PS), implants (IMP), and their interactions (PS × IMP) on primal cut weights (%; mean ± standard error).

	CF		YF		*p*-Value		
	No IMP	IMP	No IMP	IMP	PS	IMP	PS × IMP
**Round**	24.0 ± 0.65	24.1 ± 0.66	23.4 ± 0.64	23.5 ± 0.65	<0.01	0.29	0.72
**Loin**	14.7 ^a^ ± 0.14	14.8 ^a^ ± 0.14	14.5 ^b^ ± 0.14	14.8 ^a^ ± 0.14	0.10	<0.01	0.02
**Flank**	6.49 ± 0.72	6.29 ± 0.70	6.36 ± 0.71	6.28 ± 0.70	0.03	<0.01	0.18
**Chuck**	28.0 ± 0.27	28.2 ± 0.27	28.3 ± 0.27	28.7 ± 0.27	<0.01	<0.01	0.20
**Rib**	10.3 ^a^ ± 0.37	10.4 ^a^ ± 0.37	10.4 ^a^ ± 0.37	10.2 ^b^ ± 0.36	0.12	0.29	0.03
**Plate**	6.96 ± 0.53	6.80 ± 0.52	7.33 ± 0.55	7.17 ± 0.54	<0.01	<0.01	0.94
**Brisket**	5.20 ± 0.31	5.26 ± 0.32	5.35 ± 0.32	5.27 ± 0.32	0.05	0.82	0.06
**Shank**	3.83 ± 0.09	3.77 ± 0.09	3.71 ± 0.09	3.64 ± 0.09	<0.01	<0.01	0.92

^a, b^ Means with different superscripts were significantly different (*p* < 0.05): CF, calf-fed production system; YF, yearling-fed production system; IMP, implanted; No IMP, non-implanted.

**Table 4 foods-11-00518-t004:** Effects of production system (PS), implants (IMP), and their interactions (PS × IMP) on lean and fat content (%) of individual primals obtained through complete dissection (mean ± standard error).

	CF		YF		*p*-Value		
	No IMP	IMP	No IMP	IMP	PS	IMP	PS × IMP
Lean Component							
**Round**	65.0 ± 0.94	65.9 ± 0.93	65.0 ± 0.94	65.6 ± 0.94	0.47	<0.01	0.26
**Loin**	56.8 ± 1.88	57.6 ± 1.87	56.6 ± 1.88	56.8 ± 1.88	0.02	0.02	0.15
**Flank**	45.5 ± 3.60	47.2 ± 3.62	45.8 ± 3.60	47.0 ± 3.62	0.93	<0.01	0.50
**Chuck**	60.1 ± 1.33	62.1 ± 1.31	60.3 ± 1.33	61.9 ± 1.31	0.78	<0.01	0.28
**Rib**	49.0 ± 2.27	51.2 ± 2.27	48.8 ± 2.27	50.5 ± 2.27	0.09	<0.01	0.27
**Plate**	44.2 ± 2.62	46.9 ± 2.65	44.0 ± 2.62	46.3 ± 2.65	0.16	<0.01	0.47
**Brisket**	42.5 ± 1.61	44.8 ± 1.63	42.7 ± 1.61	44.5 ± 1.63	0.95	<0.01	0.34
**Shank**	43.4 ± 0.51	44.1 ± 0.52	44.6 ± 0.52	45.0 ± 0.52	<0.01	<0.01	0.24
**Total**	56.1 ± 1.73	57.7 ± 1.72	56.0 ± 1.74	57.2 ± 1.73	0.17	<0.01	0.33
**Fat component**							
**Round**	18.5 ± 1.13	17.7 ± 1.09	18.8 ± 1.15	18.5 ± 1.13	<0.01	<0.01	0.12
**Loin**	27.5 ± 2.13	26.4 ± 2.08	27.7 ± 2.14	27.3 ± 2.12	0.03	<0.01	0.18
**Flank**	53.6 ± 3.52	51.9 ± 3.54	53.3 ± 3.52	52.0 ± 3.54	0.88	<0.01	0.52
**Chuck**	26.0 ± 1.78	24.0 ± 1.69	26.0 ± 1.79	24.6 ± 1.72	0.11	<0.01	0.19
**Rib**	33.1 ± 3.42	31.0 ± 3.30	33.7 ± 3.45	32.2 ± 3.37	<0.01	<0.01	0.32
**Plate**	43.6 ± 2.47	40.6 ± 2.43	43.8 ± 2.47	41.3 ± 2.44	0.24	<0.01	0.54
**Brisket**	43.6 ± 2.99	40.5 ± 2.94	43.8 ± 3.00	41.5 ± 2.96	0.05	<0.01	0.13
**Shank**	15.5 ^a^ ± 0.57	14.8 ^b^ ± 0.55	15.3 ^a^ ± 0.57	15.5 ^a^ ± 0.58	0.05	0.05	<0.01
**Total**	28.7 ± 2.21	26.9 ± 2.13	29.0 ± 2.22	27.8 ± 2.17	0.01	<0.01	0.26

^a, b^ Means with different superscripts were significantly different (*p* < 0.05): CF, calf-fed production system; YF, yearling-fed production system; IMP, implanted; No IMP, non-implanted.

**Table 5 foods-11-00518-t005:** Coefficient of determination (R2) depicting variance explained by response surface regression models for cutout parameters under calf-fed (CF) and yearling-fed (YF) production systems.

	Primal Weight (%)		Lean Component		Fat Component	
	CF	YF	CF	YF	CF	YF
**Round**	0.35	0.58	0.24	0.76	0.27	0.55
**Loin**	0.29	0.76	0.29	0.55	0.30	0.45
**Flank**	0.43	0.57	0.31	0.36	0.31	0.37
**Chuck**	0.32	0.62	0.30	0.61	0.29	0.45
**Rib**	0.37	0.38	0.32	0.46	0.31	0.43
**Plate**	0.32	0.31	0.26	0.42	0.28	0.40
**Brisket**	0.14	0.32	0.22	0.32	0.29	0.37
**Shank**	0.28	0.93	0.40	0.79	0.28	0.57
**Total**	-	-	0.29	0.56	0.31	0.39

CF, calf-fed production system; YF, yearling-fed production system.

## Data Availability

Datasets generated during the current study are available from the corresponding author on reasonable request.

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
