# Peer review of "Influence of Production Factors on Beef Primal Tissue Composition"

_foods, 2022, doi:10.3390/foods11040518_

Round 1

Reviewer 1 Report

The authors consider an evaluation of the effects of beef production systems and implants on muscle weight and composition. 

Comments:

This is an important research topic and has relevance beyond that identified in the paper. It could help inform cut by cook eating quality prediction models, such as the Meat Standards Australia eating quality model [1]. Along these lines, the relationship between marbling across muscles has been studied in [2] and the link with eating quality has been investigated recently in [3] which considered both human grader measurements and objective fat measurements.

The experiment was nicely balanced over the two production systems and implant presence/absence. I think one of the key take home messages about this study is that there is little to no evidence of an interaction (particularly if we think about multiple correction, there's a lot of p-values presented and very few significant interactions found).

Queries:

1. I couldn't see any details about the type of implants used.

2. What is the range of marbling score? It was mentioned that it was "assessed subjectively using pictorial standards as reference points [27]" but it would be beneficial to have some detail provided in the paper to immediately assess the practical significance of the results presented in Table 2.

3. Is a random intercept an appropriate strategy for dealing with different cutout methodologies over the 10 year period? [Section 2.3]

4. In section 2.3 is says that "Models were compared to select the distribution with the lowest value of the Bayesian information criterion 148 [29]." Does this mean that variable selection was performed for each model? I'm not sure that's necessary given the relatively small number of predictors. Similarly, "Breed percentage covariates were tested for significance in the model and non-significant breed variables were removed." It generally causes no harm to leave in the breed variable - I wouldn't generally recommend removing individual breed dummy variables, however, if done properly this would be equivalent to creating an "Other" breed.

5. There's no question that hormone implants improve weight gain, but the impact on animal welfare and eating quality [5] are also important considerations that deserve mention (e.g. in the introduction).

6. Table 3 caption describes the data as "primal cut weights (%; mean ± standard error)" is it a weight or a percent, if it's a percent what is it a percent of? [I assume it's percent of total lean meat yield, but it's not explicitly stated.]

7. Table 4 Fat component Shank: it's a little surprising to see a significant interaction when the estimated marginal means are so similar, and the standard error is relatively large. Please check.

8. More generally, the statistical analysis section was vague about what predictors (potentially after model selection) and random effects were included in each specific analysis. This information should be included, for example in the caption for each table of results.

References:

1. Polkinghorne, R. J. & Thompson, J. M. Meat standards and grading: A world view. Meat Sci 86, 227–235 (2010) doi:10.1016/j.meatsci.2010.05.010.  
2. Konarska, M., Kuchida, K., Tarr, G. & Polkinghorne, R. J. Relationships between marbling measures across principal muscles. Meat Sci 123, 67–78 (2017) doi:10.1016/j.meatsci.2016.09.005.
3. Stewart, S. M. et al. Prediction of consumer palatability in beef using visual marbling scores and chemical intramuscular fat percentage. Meat Sci 108322 (2020) doi:10.1016/j.meatsci.2020.108322.
4. Packer, D. T., Geesink, G. H., Polkinghorne, R., Thompson, J. M. & Ball, A. J. The impact of two different hormonal growth promotants (HGPs) on the eating quality of feedlot-finished steer carcasses. Anim Prod Sci 59, 384 (2018) doi:10.1071/AN17121.

Reviewer 2 Report

The study aimed to evaluate the effect of production system and implants on tissue/carcass composition. The subject is interesting and the manuscript is well-structured and written. I just have few comments (attached PDF) that need to be addressed by the authors before been accepted for publication.
